# ResT: An Efficient Transformer for Visual Recognition

**Qing-Long Zhang, Yu-Bin Yang**
State Key Laboratory for Novel Software Technology
Nanjing University, Nanjing 21023, China
wofmanaf@smail.nju.edu.cn, yangyubin@nju.edu.cn

## Abstract

This paper presents an efficient multi-scale vision Transformer, called ResT, that capably served as a general-purpose backbone for image recognition. Unlike existing Transformer methods, which employ standard Transformer blocks to tackle raw images with a fixed resolution, our ResT have several advantages: (1) A memory-efficient multi-head self-attention is built, which compresses the memory by a simple depth-wise convolution, and projects the interaction across the attention-heads dimension while keeping the diversity ability of multi-heads; (2) Positional encoding is constructed as spatial attention, which is more flexible and can tackle with input images of arbitrary size without interpolation or fine-tune; (3) Instead of the straightforward tokenization at the beginning of each stage, we design the patch embedding as a stack of overlapping convolution operation with stride on the token map. We comprehensively validate ResT on image classification and downstream tasks. Experimental results show that the proposed ResT can outperform the recently state-of-the-art backbones by a large margin, demonstrating the potential of ResT as strong backbones. The code and models will be made publicly available at https://github.com/wofmanaf/ResT.

## 1 Introduction

Deep learning backbone architectures have been evolved for years and boost the performance of computer vision tasks such as classification [5, 26, 33, 11], object detection [2, 41, 18, 25], and instance segmentation [10, 24, 31], etc.

There are mainly two types of backbone architectures most commonly applied in computer vision: convolutional network (CNN) architectures [11, 38] and Transformer ones [6, 5, 33, 39]. Both of them capture feature information by stacking multiple blocks. The CNN block is generally a bottleneck structure [11], which can be defined as a stack of $1 \times 1$, $3 \times 3$, and $1 \times 1$ convolution layers with residual learning (shown in Figure 1a). The $1 \times 1$ layers are responsible for reducing and then increasing channel dimensions, leaving the $3 \times 3$ layer a bottleneck with smaller input/output channel dimensions. The CNN backbones are generally faster and require less inference time thanks to parameter sharing, local information aggregation, and dimension reduction. However, due to the limited and fixed receptive field, CNN blocks may be less effective in scenarios that require modeling long-range dependencies. For example, in instance segmentation, being able to collect and associate scene information from a large neighborhood can be useful in learning relationships across objects [23].

To overcome these limitations, Transformer backbones are recently explored for their ability to capture long-distance information [5, 33, 26, 19]. Unlike CNN backbones, the Transformer ones first split an image into a sequence of patches (i.e., tokens), then sum these tokens with positional encoding to represent coarse spatial information, and finally adopt a stack of Transformer blocks to capture feature information. A standard Transformer block [28] comprises a multi-head self-attention (MSA) that employs a query-key-value decomposition to model global relationships between sequence

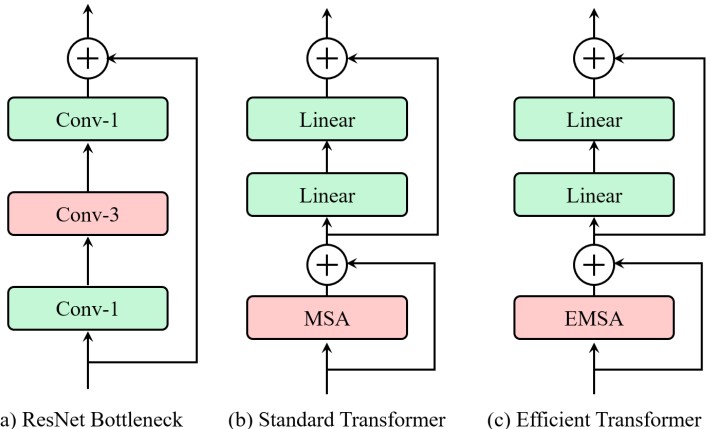

| (a) ResNet Bottleneck | (b) Standard Transformer | (c) Efficient Transformer |

Figure 1: Examples of backbone blocks. **Left:** A standard ResNet Bottleneck Block [11]. **Middle:** A Standard Transformer Block. **Right:** The proposed Efficient Transformer Block. The only difference compared with standard Transformer block is the replacement of the Multi-Head Self-Attention (MSA) with Efficient Multi-head Self-Attention (EMSA).

tokens, and a feed-forward network (FFN) to learn wider representations (shown in Figure 1b). As a result, Transformer blocks can dynamically adapt the receptive field according to the image content.

Despite showing great potential than CNNs, the Transformer backbones still have four major shortcomings: (1) It is difficult to extract the low-level features which form some fundamental structures in images (e.g., corners and edges) since existing Transformer backbones direct perform tokenization of patches from raw input images. (2) The memory and computation for MSA in Transformer blocks scale quadratically with spatial or embedding dimensions (i.e., the number of channels), causing vast overheads for training and inference. (3) Each head in MSA is responsible for only a subset of embedding dimensions, which may impair the performance of the network, particularly when the tokens embedding dimension (for each head) is short, making the dot product of query and key unable to constitute an informative function. (4) The input tokens and positional encoding in existing Transformer backbones are all of a fixed scale, which are unsuitable for vision tasks that require dense prediction.

In this paper, we proposed an efficient general-purpose backbone ResT (named after ResNet [11]) for computer vision, which can remedy the above issues. As illustrated in Figure 2, ResT shares exactly the same pipeline of ResNet, i.e., a stem module applied for extracting low-level information and strengthening locality, followed by four stages to construct hierarchical feature maps, and finally a head module for classification. Each stage consists of a patch embedding, a positional encoding module, and multiple Transformer blocks with specific spatial resolution and channel dimension. The patch embedding module creates a multi-scale pyramid of features by hierarchically expanding the channel capacity while reducing the spatial resolution with overlapping convolution operations. Unlike the conventional methods which can only tackle images with a fixed scale, our positional encoding module is constructed as spatial attention which is conditioned on the local neighborhood of the input token. By doing this, the proposed method is more flexible and can process input images of arbitrary size without interpolation or fine-tune. Besides, to improve the efficiency of the MSA, we build an efficient multi-head self-attention (EMSA), which significantly reduce the computation cost by a simple overlapping Depth-wise Conv2d. In addition, we compensate short-length limitations of the input token for each head by projecting the interaction across the attention-heads dimension while keeping the diversity ability of multi-heads.

We comprehensively validate the effectiveness of the proposed ResT on the commonly used benchmarks, including image classification on ImageNet-1k and downstream tasks, such as object detection, and instance segmentation on MS COCO2017. Experimental results demonstrate the effectiveness and generalization ability of the proposed ResT compared with the recently state-of-the-art Vision Transformers and CNNs. For example, with a similar model size as ResNet-18 (69.7%) and PVT-Tiny (75.1%), our ResT-Small obtains a Top-1 accuracy of 79.6% on ImageNet-1k.

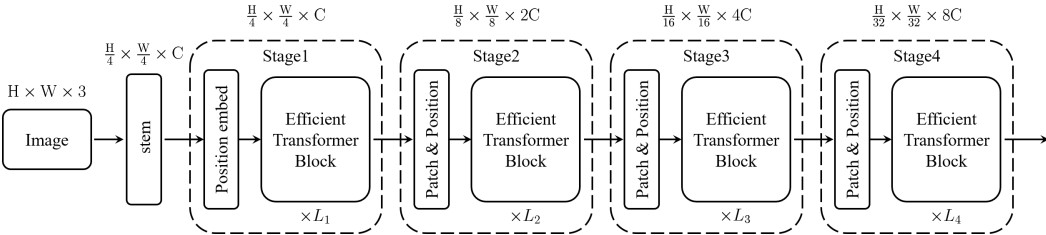

Figure 2: The pipeline of the proposed ResT. Similar to ResNet [11], ResT build stages with stacked blocks, making it flexible to serve as the backbone of downstream tasks, such as Object detection, Person ReID, and Instance Segmentation, etc.

## 2 ResT

As illustrated in Figure 2, ResT shares exactly the same pipeline as ResNet [11], i.e., a stem module applied to extract low-level information, followed by four stages to capture multi-scale feature maps. Each stage consists of three components, one patch embedding module (or stem module), one positional encoding module, and a set of $L$ efficient Transformer blocks. Specifically, at the beginning of each stage, the patch embedding module is adopted to reduce the resolution of the input token and expanding the channel dimension. The positional encoding module is fused to restrain position information and strengthen the feature extracting ability of patch embedding. After that, the input token is fed to the efficient Transformer blocks (illustrated in Figure 1c). In the following sections, we will introduce the intuition behind ResT.

### 2.1 Rethinking of Transformer Block

The standard Transformer block consists of two sub-layers of MSA and FFN. A residual connection is employed around each sub-layer. Before MSA and FFN, layer normalization (LN [1]) is applied. For a token input $x \in \mathbb{R}^{n \times d_m}$, where $n$, $d_m$ indicates the spatial dimension, channel dimension, respectively. The output for each Transformer block is:

$$y = x' + \text{FFN}(\text{LN}(x')), \text{ and } x' = x + \text{MSA}(\text{LN}(x)) \tag{1}$$

**MSA.** MSA first obtains query $\mathbf{Q}$, key $\mathbf{K}$, and value $\mathbf{V}$ by applying three sets of projections to the input, each consisting of $k$ linear layers (i.e., heads) that map the $d_m$ dimensional input into a $d_k$ dimensional space, where $d_k = d_m/k$ is the head dimension. For the convenience of description, we assume $k = 1$, then MSA can be simplified to single-head self-attention (SA). The global relationship between the token sequence can be defined as

$$\text{SA}(\mathbf{Q}, \mathbf{K}, \mathbf{V}) = \text{Softmax}(\frac{\mathbf{Q}\mathbf{K}^{\text{T}}}{\sqrt{d_k}})\mathbf{V} \tag{2}$$

The output values of each head are then concatenated and linearly projected to form the final output. The computation costs of MSA are $\mathcal{O}(2d_m n^2 + 4d_m^2 n)$, which scale quadratically with spatial dimension or embedding dimensions according to the input token.

**FFN.** The FFN is applied for feature transformation and non-linearity. It consists of two linear layers with a non-linearity activation. The first layer expands the embedding dimensions of the input from $d_m$ to $d_f$ and the second layer reduce the dimensions from $d_f$ to $d_m$.

$$\text{FFN}(x) = \sigma(x\mathbf{W}_1 + \mathbf{b}_1)\mathbf{W}_2 + \mathbf{b}_2 \tag{3}$$

where $\mathbf{W}_1 \in \mathbb{R}^{d_m \times d_f}$ and $\mathbf{W}_2 \in \mathbb{R}^{d_f \times d_m}$ are weights of the two Linear layers respectively, $\mathbf{b}_1 \in \mathbb{R}^{d_f}$ and $\mathbf{b}_2 \in \mathbb{R}^{d_m}$ are the bias terms, and $\sigma(\cdot)$ is the activation function GELU [12]. In standard Transformer block, the channel dimensions are expanded by a factor of 4, i.e., $d_f = 4d_m$. The computation costs of FFN are $8nd_m^2$.

### 2.2 Efficient Transformer Block

As analyzed above, MSA has two shortcomings: (1) The computation scales quadratically with $d_m$ or $n$ according to the input token, causing vast overheads for training and inference; (2) Each head in

MSA only responsible for a subset of embedding dimensions, which may impair the performance of the network, particularly when the tokens embedding dimension (for each head) is short.

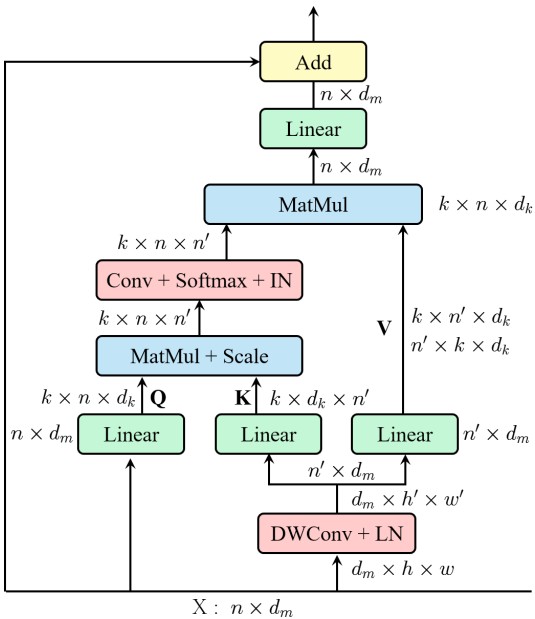

Figure 3: Efficient Multi-Head Self-Attention.

To remedy these issues, we propose an efficient multi-head self-attention module (illustrated in Figure 3). Here, we make some explanations.

(1) Similar to MSA, EMSA first adopt a set of projections to obtain query $\mathbf{Q}$.

(2) To compress memory, the 2D input token $\mathrm{x} \in \mathbb{R}^{n \times d_m}$ is reshaped to 3D one along the spatial dimension (i.e., $\hat{\mathrm{x}} \in \mathbb{R}^{d_m \times h \times w}$) and then feed to a depth-wise convolution operation to reduce the height and width dimension by a factor $s$. To make simple, $s$ is adaptive set by the feature map size or the stage number. The kernel size, stride and padding are $s + 1$, $s$, and $s/2$ respectively.

(3) The new token map after spatial reduction $\hat{\mathrm{x}} \in \mathbb{R}^{d_m \times h/s \times w/s}$ is then reshaped to 2D one, i.e., $\hat{\mathrm{x}} \in \mathbb{R}^{n' \times d_m}$, $n' = h/s \times w/s$. Then $\hat{\mathrm{x}}$ is feed to two sets of projection to get key $\mathbf{K}$ and value $\mathbf{V}$.

(4) After that, we adopt Eq. 4 to compute the attention function on query $\mathbf{Q}$, $\mathbf{K}$ and value $\mathbf{V}$.

$$\mathrm{EMSA}(\mathbf{Q}, \mathbf{K}, \mathbf{V}) = \mathrm{IN}(\mathrm{Softmax}(\mathrm{Conv}(\frac{\mathbf{Q}\mathbf{K}^{\mathrm{T}}}{\sqrt{d_k}})))\mathbf{V} \tag{4}$$

Here, $\mathrm{Conv}(\cdot)$ is a standard $1 \times 1$ convolutional operation, which model the interactions among different heads. As a result, attention function of each head can depend on all of the keys and queries. However, this will impair the ability of MSA to jointly attend to information from different representation subsets at different positions. To restore this diversity ability, we add an Instance Normalization [27] (i.e, $\mathrm{IN}(\cdot)$) for the dot product matrix (after Softmax).

(5) Finally, the output values of each head are then concatenated and linearly projected to form the final output.

The computation costs of EMSA are $\mathcal{O}(\frac{2d_m n^2}{s^2} + 2d_m^2 n(1 + \frac{1}{s^2}) + d_m n \frac{(s+1)^2}{s^2} + \frac{k^2 n^2}{s^2})$, much lower than the original MSA (assume $s > 1$), particularly in lower stages, where $s$ is tend to higher.

Also, we add FFN after EMSA for feature transformation and non-linearity. The output for each efficient Transformer block is:

$$\mathrm{y} = \mathrm{x}' + \mathrm{FFN}(\mathrm{LN}(\mathrm{x}')), \text{ and } \mathrm{x}' = \mathrm{x} + \mathrm{EMSA}(\mathrm{LN}(\mathrm{x})) \tag{5}$$

## 2.3 Patch Embedding

The standard Transformer receives a sequence of token embeddings as input. Take ViT [5] as an example, the input image $x \in \mathbb{R}^{3 \times h \times w}$ is split with a patch size of $p \times p$. These patches are flattened into 2D ones and then mapped to latent embeddings with a size of $c$, i.e, $x \in \mathbb{R}^{n \times c}$, where $n = hw/p^2$. However, this straightforward tokenization is failed to capture low-level feature information (such as edges and corners) [33]. In addition, the length of tokens in ViT are all of a fixed size in different blocks, making it unsuitable for downstream vision tasks such as object detection and instance segmentation that require multi-scale feature map representations.

Here, we build an efficient multi-scale backbone, calling ResT, for dense prediction. As introduced above, the efficient Transformer block in each stage operates on the same scale with identical resolution across the channel and spatial dimensions. Therefore, the patch embedding modules are required to progressively expand the channel dimension, while simultaneously reducing the spatial resolution throughout the network.

Similar to ResNet, the stem module (can be seen as the first patch embedding module) are adopted to shrunk both the height and width dimension with a reduction factor of 4. To effectively capture the low-feature information with few parameters, here we introduce a simple but effective way, i.e, stacking three $3 \times 3$ standard convolution layers (all with padding 1) with stride 2, stride 1, and stride 2, respectively. Batch Normalization [14] and ReLU activation [7] are applied for the first two layers. In stage 2, stage 3, and stage 4, the patch embedding module is adopted to down-sample the spatial dimension by $4 \times$ and increase the channel dimension by $2 \times$. This can be done by a standard $3 \times 3$ convolution with stride 2 and padding 1. For example, patch embedding module in stage 2 changes resolution from $h/4 \times w/4 \times c$ to $h/8 \times w/8 \times 2c$ (shown in Figure 2).

## 2.4 Positional Encoding

Positional encodings are crucial to exploiting the order of sequence. In ViT [5], a set of learnable parameters are added into the input tokens to encode positions. Let $x \in \mathbb{R}^{n \times c}$ be the input, $\theta \in \mathbb{R}^{n \times c}$ be position parameters, then the encoded input can be represent as

$$\hat{x} = x + \theta \tag{6}$$

However, the length of positions is exactly the same as the input tokens length, which limits the application scenarios.

To remedy this issue, the new positional encodings are required to have variable lengths according to input tokens. Let us look closer to Eq. 6, the summation operation is much like assigning pixel-wise weights to the input. Assume $\theta$ is related with x, i.e., $\theta = \mathrm{GL}(x)$, where $\mathrm{GL}(\cdot)$ is the group linear operation with the group number of $c$. Then Eq. 6 can be modified to

$$\hat{x} = x + \mathrm{GL}(x) \tag{7}$$

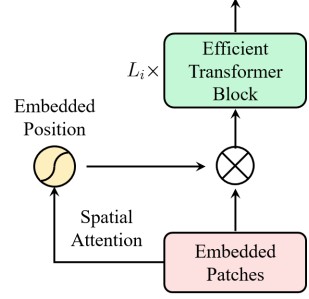

Figure 4: Patch and PE in ResT.

Besides Eq. 7, $\theta$ can also be obtained by more flexible spatial attention mechanisms. Here, we propose a simple yet effective spatial attention module calling PA(pixel-attention) to encode positions. Specifically, PA applies a $3 \times 3$ depth-wise convolution (with padding 1) operation to get the pixel-wise weight and then scaled by a sigmoid function $\sigma(\cdot)$. The positional encoding with PA module can then be represented as

$$\hat{x} = \mathrm{PA}(x) = x * \sigma(\mathrm{DWConv}(x)) \tag{8}$$

Since the input token in each stage is also obtained by a convolution operation, we can embed the positional encoding into the patch embedding module. The whole structure of stage $i$ can be illustrated in Figure 4. Note that PA can be replaced by any spatial attention modules, making the positional encoding flexible in ResT.

## 2.5 Classification Head

The classification head is performed by a global average pooling layer on the output feature map of the last stage, followed by a linear classifier. The detailed ResT architecture for ImageNet-1k is shown

in Table 1, which contains four models, i.e., ResT-Lite, ResT-Small and ResT-Base and ResT-Large, which are bench-marked to ResNet-18, ResNet-18, ResNet-50, and ResNet-101, respectively.

Table 1: Architectures for ImageNet-1k. Here, we make some definitions. "$\mathrm{Conv} - k\_c\_s$" means convolution layers with kernel size $k$, output channel $c$ and stride $s$. "MLP_$c$" is the FFN structure with hidden channel $4c$ and output channel $c$. And "EMSA_$n\_r$" is the EMSA operation with the number of heads $n$ and reduction $r$. "C" is 64 for ResT-Lite and ResT-Small, and 96 for ResT-Base and ResT-Large."PA" is short for pixel-wise attention, which are introduced in Section 2.4.

| Name | Output | Lite | Small | Base | Large |
|---|---|---|---|---|---|
| stem | $56\times56$ | patch_embed: Conv-3_C/2_2, Conv-3_C/2_1, Conv-3_C_2,PA | | | |
| stage1 | $56\times56$ | $\begin{bmatrix}\text{EMSA\_1\_8}\\\text{MLP\_64}\end{bmatrix}\times2$ | $\begin{bmatrix}\text{EMSA\_1\_8}\\\text{MLP\_64}\end{bmatrix}\times2$ | $\begin{bmatrix}\text{EMSA\_1\_8}\\\text{MLP\_96}\end{bmatrix}\times2$ | $\begin{bmatrix}\text{EMSA\_1\_8}\\\text{MLP\_96}\end{bmatrix}\times2$ |
| stage2 | $28\times28$ | patch_embed: Conv-3_2C_2, PA | | | |
| | | $\begin{bmatrix}\text{EMSA\_2\_4}\\\text{MLP\_128}\end{bmatrix}\times2$ | $\begin{bmatrix}\text{EMSA\_2\_4}\\\text{MLP\_128}\end{bmatrix}\times2$ | $\begin{bmatrix}\text{EMSA\_2\_4}\\\text{MLP\_192}\end{bmatrix}\times2$ | $\begin{bmatrix}\text{EMSA\_2\_4}\\\text{MLP\_192}\end{bmatrix}\times2$ |
| stage3 | $14\times14$ | patch_embed: Conv-3_4C_2, PA | | | |
| | | $\begin{bmatrix}\text{EMSA\_4\_2}\\\text{MLP\_256}\end{bmatrix}\times2$ | $\begin{bmatrix}\text{EMSA\_4\_2}\\\text{MLP\_256}\end{bmatrix}\times6$ | $\begin{bmatrix}\text{EMSA\_4\_2}\\\text{MLP\_384}\end{bmatrix}\times6$ | $\begin{bmatrix}\text{EMSA\_4\_2}\\\text{MLP\_384}\end{bmatrix}\times18$ |
| stage4 | $7\times7$ | patch_embed: Conv-3_8C_2, PA | | | |
| | | $\begin{bmatrix}\text{EMSA\_8\_1}\\\text{MLP\_512}\end{bmatrix}\times2$ | $\begin{bmatrix}\text{EMSA\_8\_1}\\\text{MLP\_512}\end{bmatrix}\times2$ | $\begin{bmatrix}\text{EMSA\_8\_1}\\\text{MLP\_768}\end{bmatrix}\times2$ | $\begin{bmatrix}\text{EMSA\_8\_1}\\\text{MLP\_768}\end{bmatrix}\times2$ |
| Classifier | $1\times1$ | average pool, 1000d fully-connected | | | |
| GFLOPs | | 1.4 | 1.94 | 4.26 | 7.91 |

# 3 Experiments

In this section, we conduct experiments on common-used benchmarks, including ImageNet-1k for classification, MS COCO2017 for object detection, and instance segmentation. In the following subsections, we first compared the proposed ResT with the previous state-of-the-arts on the three tasks. Then we adopt ablation studies to validate the important design elements of ResT.

## 3.1 Image Classification on ImageNet-1k

**Settings.** For image classification, we benchmark the proposed ResT on ImageNet-1k, which contains 1.28M training images and 50k validation images from 1,000 classes. The setting mostly follows [26]. Specifically, we employ the AdamW [20] optimizer for 300 epochs using a cosine decay learning rate scheduler and 5 epochs of linear warm-up. A batch size of 2048 (using 8 GPUs with 256 images per GPU), an initial learning rate of 5e-4, a weight decay of 0.05, and gradient clipping with a max norm of 5 are used. We include most of the augmentation and regularization strategies of [26] in training, including RandAugment [4], Mixup [35], Cutmix [34], Random erasing [40], and stochastic depth [13]. An increasing degree of stochastic depth augmentation is employed for larger models, i.e., 0.1, 0.1, 0.2, 0.3 for ResT-Lite, Rest-Small, ResT-Base, and ResT-Large, respectively. For the testing on the validation set, the shorter side of an input image is first resized to 256, and a center crop of 224 × 224 is used for evaluation.

**Results.** Table 2 presents comparisons to other backbones, including both Transformer-based ones and ConvNet-based ones. We can see, compared to the previous state-of-the-art Transformer-based architectures with similar model complexity, the proposed ResT achieves significant improvement by a large margin. For example, for smaller models, ResT noticeably surpass the counterpart PVT architectures with similar complexities: +4.5% for ResT-Small (79.6%) over PVT-T (75.1%). For larger models, ResT also significantly outperform the counterpart Swin architectures with similar

Table 2: Comparison with state-of-the-art backbones on ImageNet-1k benchmark. Throughput (images / s) is measured on a single V100 GPU, following [26]. All models are trained and evaluated on 224×224 resolution. The best records and the improvements over bench-marked ResNets are marked in **bold** and blue, respectively.

| Model | #Params (M) | FLOPs (G) | Throughput | Top-1 (%) | Top-5 (%) |
|---|---|---|---|---|---|
| ConvNet | | | | | |
| ResNet-18 [11] | 11.7 | 1.8 | 1852 | 69.7 | 89.1 |
| ResNet-50 [11] | 25.6 | 4.1 | 871 | 79.0 | 94.4 |
| ResNet-101 [11] | 44.7 | 7.9 | 635 | 80.3 | 95.2 |
| RegNetY-4G [22] | 20.6 | 4.0 | 1156 | 79.4 | 94.7 |
| RegNetY-8G [22] | 39.2 | 8.0 | 591 | 79.9 | 94.9 |
| RegNetY-16G [22] | 83.6 | 15.9 | 334 | 80.4 | 95.1 |
| Transformer | | | | | |
| DeiT-S [26] | 22.1 | 4.6 | 940 | 79.8 | 94.9 |
| DeiT-B [26] | 86.6 | 17.6 | 292 | 81.8 | 95.6 |
| PVT-T [29] | 13.2 | 1.9 | 1038 | 75.1 | 92.4 |
| PVT-S [29] | 24.5 | 3.7 | 820 | 79.8 | 94.9 |
| PVT-M [29] | 44.2 | 6.4 | 526 | 81.2 | 95.6 |
| PVT-L [29] | 61.4 | 9.5 | 367 | 81.7 | 95.9 |
| Swin-T [19] | 28.29 | 4.5 | 755 | 81.3 | 95.5 |
| Swin-S [19] | 49.61 | 8.7 | 437 | 83.3 | 96.2 |
| Swin-B [19] | 87.77 | 15.4 | 278 | 83.5 | 96.5 |
| **ResT-Lite (Ours)** | 10.49 | 1.4 | 1246 | **77.2** (↑ 7.5) | **93.7** (↑ 4.6) |
| **ResT-Small (Ours)** | 13.66 | 1.9 | 1043 | **79.6** (↑ 9.9) | **94.9** (↑ 5.8) |
| **ResT-Base (Ours)** | 30.28 | 4.3 | 673 | **81.6** (↑ 2.6) | **95.7** (↑ 1.3) |
| **ResT-Large (Ours)** | 51.63 | 7.9 | 429 | **83.6** (↑ 3.3) | **96.3** (↑ 1.1) |

complexities: +0.3% for ResT-Base (81.6%) over Swin-T (81.3%), and +0.3% for ResT-Large (83.6%) over Swin-S(83.3%) using $224 \times 224$ input.

Compared with the state-of-the-art ConvNets, i.e., RegNet, the ResT with similar model complexity also achieves better performance: an average improvement of 1.7% in terms of Top-1 Accuracy. Note that RegNet is trained via thorough architecture search, the proposed ResT is adapted from the standard Transformer and has strong potential for further improvement.

### 3.2 Object Detection and Instance Segmentation on COCO

**Settings.** Object detection and instance segmentation experiments are conducted on COCO 2017, which contains 118k training, 5k validation, and 20k test-dev images. We evaluate the performance of ResT using two representative frameworks: RetinaNet [18] and Mask RCNN [10]. For these two frameworks, we utilize the same settings: multi-scale training (resizing the input such that the shorter side is between 480 and 800 while the longer side is at most 1333), AdamW [20] optimizer (initial learning rate of 1e-4, weight decay of 0.05, and batch size of 16), and $1\times$ schedule (12 epochs). Unlike CNN backbones, which adopt post normalization and can directly apply to downstream tasks. ResT employs the pre-normalization strategy to accelerate network convergence, which means the output of each stage is not normalized before feeding to FPN [17]. Here, we add a layer normalization (LN [1]) for the output of each stage (before FPN [17]), similar to Swin [19]. Results are reported on the validation split.

**Object Detection Results.** Table 3 lists the results of RetinaNet with different backbones. From these results, it can be seen that for smaller models, ResT-Small is +3.6 box AP higher (40.3 vs. 36.7) than PVT-T with a similar computation cost. For larger models, our ResT-Base surpassing the PVT-S by +1.6 box AP.

Table 3: Object detection performance on the COCO val2017 split using the RetinaNet framework.

| Backbones | AP50:95 | AP50 | AP75 | APs | APm | APl | Param (M) |
|---|---|---|---|---|---|---|---|
| R18 [11] | 31.8 | 49.6 | 33.6 | 16.3 | 34.3 | 43.2 | 21.3 |
| PVT-T [29] | 36.7 | 56.9 | 38.9 | 22.6 | 38.8 | 50.0 | 23.0 |
| **ResT-Small(Ours)** | **40.3** | 61.3 | 42.7 | 25.7 | 43.7 | 51.2 | 23.4 |
| R50 [11] | 37.4 | 56.7 | 40.3 | 23.1 | 41.6 | 48.3 | 37.9 |
| PVT-S [29] | 40.4 | 61.3 | 43.0 | 25.0 | 42.9 | 55.7 | 34.2 |
| Swin-T [19] | 41.5 | 62.1 | 44.1 | 27.0 | 44.2 | 53.2 | 38.5 |
| **ResT-Base (Ours)** | **42.0** | 63.2 | 44.8 | 29.1 | 45.3 | 53.3 | 40.5 |
| R101 [11] | 38.5 | 57.8 | 41.2 | 21.4 | 42.6 | 51.1 | 56.9 |
| PVT-M [29] | 41.9 | 63.1 | 44.3 | 25.0 | 44.9 | 57.6 | 53.9 |
| Swin-S [19] | 44.5 | 65.7 | 47.5 | 27.4 | 48.0 | 59.9 | 59.8 |
| **ResT-Large (Ours)** | **44.8** | 66.1 | 48.0 | 28.3 | 48.7 | 60.3 | 61.8 |

**Instance Segmentation Results.** Table 4 compares the results of ResT with those of previous state-of-the-art models on the Mask RCNN framework. Rest-Small exceeds PVT-T by +2.9 box AP and +2.1 mask AP on the COCO val2017 split. As for larger models, ResT-Base brings consistent +1.2 and +0.9 gains over PVT-S in terms of box AP and mask AP, with slightly larger model size.

Table 4: Object detection and instance segmentation performance on the COCO val2017 split using Mask RCNN framework.

| Backbones | $AP^{box}$ | $AP^{box}_{50}$ | $AP^{box}_{75}$ | $AP^{mask}$ | $AP^{mask}_{50}$ | $AP^{mask}_{75}$ | Param (M) |
|---|---|---|---|---|---|---|---|
| R18 [11] | 34.0 | 54.0 | 36.7 | 31.2 | 51.0 | 32.7 | 31.2 |
| PVT-T [29] | 36.7 | 59.2 | 39.3 | 35.1 | 56.7 | 37.3 | 32.9 |
| **ResT-Small(Ours)** | **39.6** | 62.9 | 42.3 | **37.2** | 59.8 | 39.7 | 33.3 |
| R50 [11] | 38.6 | 59.5 | 42.1 | 35.2 | 56.3 | 37.5 | 44.3 |
| PVT-S [29] | 40.4 | 62.9 | 43.8 | 37.8 | 60.1 | 40.3 | 44.1 |
| **ResT-Base(Ours)** | **41.6** | 64.9 | 45.1 | **38.7** | 61.6 | 41.4 | 49.8 |

## 3.3 Ablation Study

In this section, we report the ablation studies of the proposed ResT, using ImageNet-1k image classification. To thoroughly investigate the important design elements, we only adopt the simplest data augmentation and hyper-parameters settings in [11]. Specifically, the input images are randomly cropped to $224 \times 224$ with random horizontal flipping. All the architectures of ResT-Lite are trained with SGD optimizer (with weight decay 1e-4 and momentum 0.9) for 100 epochs, starting from the initial learning rate of $0.1 \times \text{batch\_size}/512$ (with a linear warm-up of 5 epochs) and decreasing it by a factor of 10 every 30 epochs. Also, a batch size of 2048 (using 8 GPUs with 256 images per GPU) is used.

**Different types of stem module.** Here, we test three type of stem modules: (1) the first patch embedding module in PVT [29], i.e., $4 \times 4$ convolution operation with stride 4 and no padding; (2) the stem module in ResNet [11], i.e., one $7 \times 7$ convolution layer with stride 2 and padding 3, followed by one $3 \times 3$ max-pooling layer; (3) the stem module in the proposed ResT, i.e., three $3 \times 3$ convolutional layers (all with padding 1) with stride 2, stride 1, and stride 2, respectively. We report the results in Table 5. The stem module in the proposed ResT is more effective than that in PVT and ResNet: +0.92% and +0.64% improvements in terms of Top-1 accuracy, respectively.

**Ablation study on EMSA.** As shown in Figure!3, we adopt a Depth-wise Conv2d to reduce the computation of MSA. Here, we provide the comparison of more strategies with the same reduction stride $s$. Results are shown in Table 6. As can be seen, average pooling achieves slightly worse

Table 5: Comparison of various stem modules on ResT-Lite. Results show that the proposed stem module is more effective than existing ones in PVT and ResNet.

| Stem | Top-1 (%) | Top-5 (%) |
|------|-----------|-----------|
| PVT [29] | 71.96 | 89.87 |
| ResNet [11] | 72.24 | 90.17 |
| ResT (Ours) | 72.88 | 90.62 |

Table 6: Comparison of different reduction strategies of EMSA on ResT-Lite. Results show that Average Pooling can be an alternative to Depth-wise Conv2d to make a trade-off.

| Reduction | Top-1 (%) | Top-5 (%) |
|-----------|-----------|-----------|
| DWConv | 72.88 | 90.62 |
| Avg Pooling | 72.64 | 90.41 |
| Max Pooling | 72.20 | 89.97 |

results (-0.24%) compared with the original Depth-wise Conv2d, while the results of the Max Pooling strategy are the worst. Since the pooling operation introduces no extra parameters, therefore, average pooling can be an alternative to Depth-wise Conv2d in practice.

Table 7: Ablation study results on the important design elements of EMSA on ResT-Lite, including the $1 \times 1$ convolution operation and Instance Normalization in Eq. 4.

| Methods | Top-1 (%) | Top-5 (%) |
|---------|-----------|-----------|
| origin | 72.88 | 90.62 |
| w/o IN | 71.98 | 90.32 |
| w/o Conv-1&IN | 71.72 | 89.93 |

Table 8: Comparison of various positional encoding (PE) strategies on ResT-Lite.

| Encoding | Top-1 (%) | Top-5 (%) |
|----------|-----------|-----------|
| w/o position | 71.54 | 89.82 |
| + LE | 71.98 | 90.32 |
| + GL | 72.04 | 90.41 |
| + PA | 72.88 | 90.62 |

In addition, EMSA also adding two important elements to the standard MSA, i.e., one $1 \times 1$ convolution operation to model the interaction among different heads, and the Instance Normalization(IN) to restore diversity of different heads. Here, we validate the effectiveness of these two settings. Results are shown in Table 7. We can see, without IN, the Top-1 accuracy is degraded by 0.9%, we attribute it to the destroying of diversity among different heads because the $1 \times 1$ convolution operation makes all heads focus on all the tokens. In addition, the performance drops 1.16% without the convolution operation and IN. This can demonstrate that the combination of long sequence and diversity are both important for attention function.

**Different types of positional encoding.** In section 2.4, we introduced 3 types of positional encoding types, i.e., the original learnable parameters with fixed lengths [5] (LE), the proposed group linear mode(GL), and PA mode. These encodings are added/multiplied to the input patch token at the beginning of each stage. Here, we compared the proposed GL and PA with LE, results are shown in Table 8. We can see, the Top-1 accuracy degrades from 72.88% to 71.54% when the PA encoding is removed, this means that positional encoding is crucial for ResT. The LE and GL, achieve similar performance, which means it is possible to construct variable length of positional encoding. Moreover, the PA mode significantly surpasses the GL, achieving 0.84% Top-1 accuracy improvement, which indicates that spatial attention can also be modeled as positional encoding.

## 4   Conclusion

In this paper, we proposed ResT, a new version of multi-scale Transformer which produces hierarchical feature representations for dense prediction. We compressed the memory of standard MSA and model the interaction between multi-heads while keeping the diversity ability. To tackle input images with arbitrary, we further redesign the positional encoding as spatial attention. Experimental results demonstrate that the potential of ResT as strong backbones for dense prediction. We hope that our approach will foster further research in visual recognition.

## Acknowledgments and Disclosure of Funding

This work is funded by the Natural Science Foundation of China (No. 62176119) and the program B for Outstanding PhD candidate of Nanjing University.

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
