# ResT: An Efficient Transformer for Visual Recognition

**Qing-Long Zhang, Yu-Bin Yang**[*]
State Key Laboratory for Novel Software Technology
Nanjing University, Nanjing 21023, China
`wofmanaf@smail.nju.edu.cn, yangyubin@nju.edu.cn`

## A  Appendix

We provide the related work and more experimental results to complete the experimental sections of the main paper.

### A.1  Related Work

**Convolutional Networks.** As the cornerstone of deep learning computer vision, CNNs have been evolved for years and are becoming more accurate and faster. Among them, the ResNet series [8, 20, 22] are the most famous backbone networks because of their simple design and high performance. The base structure of ResNet is the residual bottleneck, which can be defined as a stack of one $1 \times 1$, one $3 \times 3$, and one $1 \times 1$ Convolution layer with residual learning. Recent works explore replacing the $3 \times 3$ Convolution layer with more complex modules [15, 10] or combining with attention modules [24, 19]. Similar to vision Transformers, CNNs can also capture long-range dependencies if correctly incorporated with self-attention such as Non-Local or MSA. These studies show that the advantage of CNN lies in parameter sharing and focuses on the aggregation of local information, while the advantage of self-attention lies in the global receptive field and focuses on the aggregation of global information. Intuitively speaking, global and local information aggregation are both useful for vision tasks. Effectively combining global information aggregation and local information aggregation may be the right direction for designing the best network architecture.

**Vision Transformers.** Transformer is a type of neural network that mainly relies on self-attention to draw global dependencies between input and output. Recently, Transformer-based models are explored to solve various computer vision tasks such as image processing [3], classification [5, 4, 21], and object detection [2, 25], etc. Here, we focus on investigating the classification vision Transformers. These Transformers usually view an image as a sequence of patches and perform classification with a Transformer encoder. The encoder consists of several Transformer blocks, each including an MSA and an FFN. Layer-norm (LN) is applied before each layer and residual connections are employed in both the self-attention and FFN module.

Among them, ViT [4] is the first fully Transformer classification model. In particular, ViT split each image into $14 \times 14$ or $16 \times 16$ with a fixed length, then several Transformer layers are adopted to model global relation among these tokens for input classification. DeiT [16] explores the data-efficient training and distillation of ViT. Tokens-to-Tokens (T2T-ViT) [21] point out that the simple tokenization of input images in ViT fails to model the important local structure (e.g., edges, lines) among neighboring pixels, leading to its low training sample efficiency. Transformer-in-Transformer (TNT) [6] split each image into a sequence of patches and each patch is reshaped to pixel sequence. After embedding, two Transformer layers are applied for representation learning where the outer Transformer layer models the global relationships among the patch embedding and the inner one extracts local structure information of pixel embedding. Pyramid Vision Transformer (PVT) [18] follows the ResNet paradigm to construct Transformer backbones, making it more suitable for downstream tasks. MViT [7] further apply pooling function to reduce computation costs.

---

[*]Corresponding author.

35th Conference on Neural Information Processing Systems (NeurIPS 2021).

**Positional Encoding.** Different from CNNs, which can implicitly encode spatial position information by zero-padding [9], the self-attention in Transformers has no ability to distinguish token order in different positions. Therefore, positional encoding is essential for the patch embeddings to retain positional information. There are mainly two types of positional encoding most commonly applied in vision Transformers, i.e., absolute positional encoding and relative positional encoding. The absolute positional encoding is used in ViT [4] and its extended methods, where a standard learnable 1D position embedding is added to the sequence of embedded patches. The relative method is used in BoTNet [15] and Swin Transformer [13], where the split 2D relative position embeddings are added. Generally speaking, the relative positional encodings are better suited for vision tasks, this can be attributed to attention not only taking into account the content information but also relative distances between features at different locations [14].

## A.2  Visualization and Interpretation

**Analysis on EMSA.** In this part, we measure the diversity of EMSA. To simplify, we apply $\mathbf{A} \in \mathbb{R}^{k \times n \times n'}$ to denote the attention map, with $k$ be the number of heads in EMSA. Assume $\mathbf{A}_i \in \mathbb{R}^{1 \times m}$ is the $i$-th attention map (after reshape), where $m = nn'$. Then we can compute the cross-layer similarity to measure the diversity of different heads.

$$\mathrm{M}_{ij} = \frac{\mathbf{A}_i \mathbf{A}_j^{\mathrm{T}}}{\|\mathbf{A}_i\| \|\mathbf{A}_j\|} \tag{1}$$

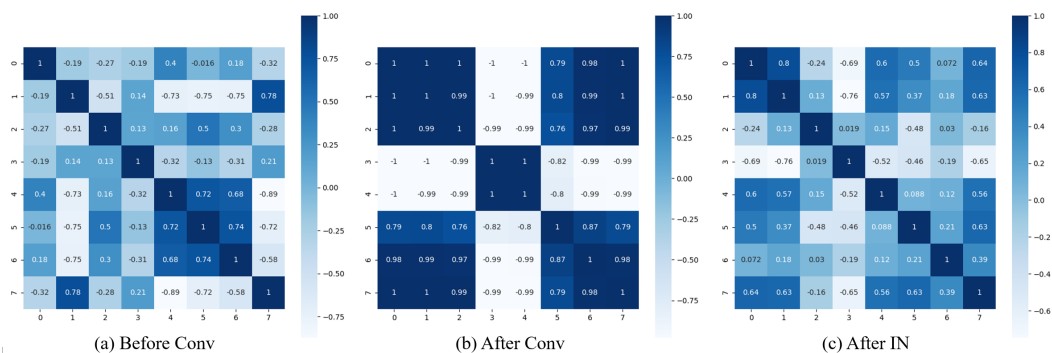

(a) Before Conv        (b) After Conv        (c) After IN

Figure 1: Attention map visualization of the last blocks of stage 4 of the ResT-Lite.

To thoroughly measure the diversity, we calculate extract three types of $\mathbf{A}$: the attention map before Conv-1 (i.e., $\mathbf{QK}^{\mathrm{T}}/\sqrt{d_k}$), the one after Conv-1 (i.e., $\mathrm{Conv}(\mathbf{QK}^{\mathrm{T}}/\sqrt{d_k})$), and the one after Instance Normalization [17] (i.e., $\mathrm{IN}(\mathrm{Softmax}(\mathrm{Conv}(\mathbf{QK}^{\mathrm{T}}/\sqrt{d_k}))))$. We randomly sample 1,000 images from the ImageNet-1k validation set and visualize the mean diversity in Figure 1.

As shown in Figure 1b, although the $1 \times 1$ convolution model the interactions among different heads, it impairs the ability of MSA to jointly attend to information from different representation subsets at different positions. After instance normalization (in Figure 1c), the diversity ability is restored.

**Interpretabilty.** In order to validate the effectiveness of ResT more intuitively, we sample 6 images from the ImageNet-1k validation split. We use Group-CAM [23] to visualize the heatmaps at the last convolutional layer of ResT-Lite. For comparison, we also draw the heatmaps of its counterpart ResNet-50. As shown in Figure 2, the proposed ResT-Lite can adaptively produce heatmaps according to the image content.

## A.3  More Experiments

**Comparisons of EMSA and MSA.** In this part, we make a quantitative comparison of the performance and efficiency of the two modules on ResT-Lite (keeping other components intact). Experimental settings are the same as Ablation Study (Section **??**7), except for the batch size of MSA, which is 512 since each V100 GPU can only tackle 64 images at the same time. We report the results

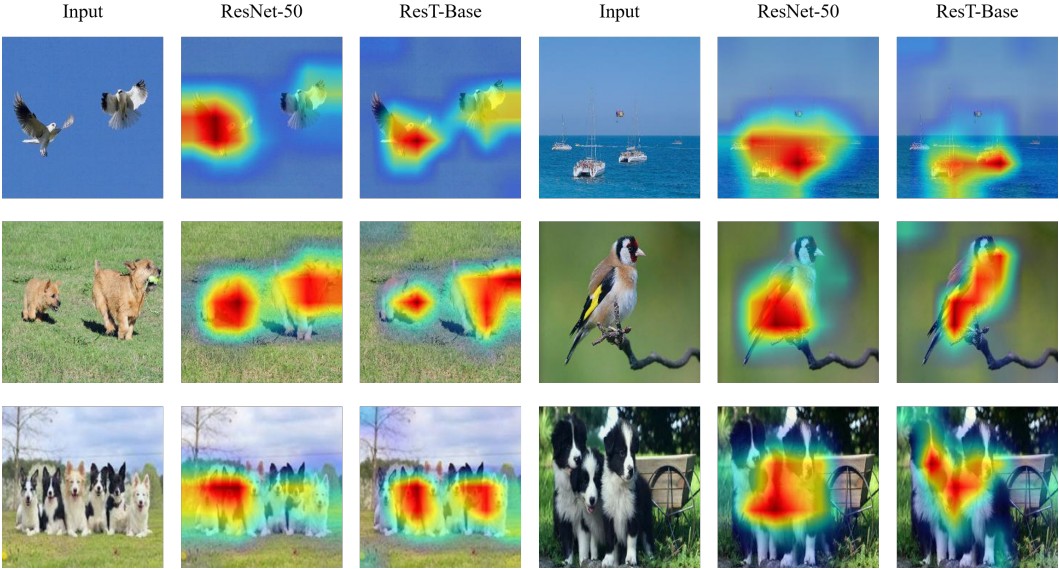

| Input | ResNet-50 | ResT-Base | Input | ResNet-50 | ResT-Base |

Figure 2: Sample visualization on ImageNet-1k val split generated by Group-CAM [23].

in Table 1. Both versions of ResT-Lite share almost the same parameters. However, the EMSA version achieves better Top-1 accuracy (+0.2%) with fewer computations(-0.2G). The actual inference throughput indicates EMSA (1246) is 2.4x faster than the original MSA(512). Therefore, EMSA can capably serve as an effective replacement for MSA.

Table 1: Comparison of MSA and EMSA.

| Model | #Params (M) | FLOPs (G) | Throughput | Top-1 (%) | Top-5 (%) |
|-------|-------------|-----------|------------|-----------|-----------|
| MSA   | 10.48       | 1.6       | 512        | 72.68     | 90.46     |
| EMSA  | 10.49       | 1.4       | 1246       | 72.88     | 90.62     |

**Object Detection.** In Section 3.2, we replace the backbone in RetinaNet [12] with ResT and add a layer normalization (LN [1]) for the output of each stage (before FPN [11]), just like Swin. Here, we further validate the effectiveness of LN. We follow the same setting as Section **??**. Results are reported in Table 2.

Table 2: Object Detection.

| Backbones | Setting | AP50:95 |
|-----------|---------|---------|
| ResT-Small | w/o LN | 39.5 |
|            | w LN   | 40.3 |
| ResT-Base  | w/o LN | 41.2 |
|            | w LN   | 42.0 |

From Table 2, we can see, LN is indeed matters in downstream tasks. An average +0.8 box AP improvement is achieved with LN for RetinaNet [12].

## A.4 Discussions

**Mathematical Definition of GL.** Given $x$ with size $\mathbb{R}^{n \times d_m}$, where $n$ is spatial dimension and $d_m$ is the channel dimension, GL first splits $x$ into $g$ non-overlapping groups,

$$x = Concat(x_1, \cdots, x_g) \qquad (2)$$

where the size of $x_i$ is $n \times \frac{d_m}{g}$.

All $x_i$'s are then simultaneously transformed by $g$ linear operations to produce $g$ outputs

$$y_i = x_i W_i \tag{3}$$

where $W_i$ is the linear operation weight.

$y_i$'s are then concatenated to produce the final $n \times d_m$ output $y = Concat(y1, \cdots, y_g)$. In ResT, we set $g = d_m$, i.e., the channel dimension of the input.

**Ablation Study Settings.** Note that the settings between ablation study and the main results are different. Here, we give the explanations. We adopt the simplest data augmentation and hyper-parameters settings in ResNet [8] to thoroughly investigate the important components of ResT, i.e., eliminating the influence of strong data augmentation and training tricks. Under this setting, we demonstrate that Vision Transformers can still achieve better results without training tricks. Specifically, the Top-1 accuracy of ResT-Lite is 72.88, which outperforms the ResNet-18 (69.76) by +3.1 improvement. We believe the same setting as ResNet in the ablation study can eliminate the doubt of Vision Transformer to some extent, i.e., the improvements of Vision Transformer over CNN mainly come with strong data augmentation and training tricks. This can promote the ongoing and future research of Vision Transformer.