# OpenReview forum: "ResT: An Efficient Transformer for Visual Recognition"
_NeurIPS.cc/2021/Conference — NeurIPS 2021 Poster_

### Official Review · Reviewer_WKZ1 · 2021-07-14

**Rating:** 5
**Confidence:** 4

**Summary:**

In this paper, the authors propose ResT, which is an efficient multi-scale vision Transformer backbone. Comparing with ViT, ResT has the following three modifications: 1) A DWconv-based efficient self-attention. 2) A DWconv-based position encoding. 3) Overlapped token embedding. The experiment results show that ResT outperforms previous methods.

**Limitations And Societal Impact:**

The authors did not point out the limitations and potential negative societal impact of their work.

**Main Review:**

Strengths
1) The paper is well written and easy to follow.
2) The results seem good.
3) The idea of using Instance Normalization after softmax is interesting and its corresponding ablation study results seem good.

Weaknesses
1) The analysis about the second shortcoming of ViT is weird. The authors think the multi-head with a low subset head dimension may influence the performance. But no experimental results are provided to prove it. It also contradicts with common sense that more head brings better model capability. Meanwhile, if the low dimension influences the performance, why not reduce the head number directly?
2) The definition of s=8/k is weird and does not make sense. Why the spatial reduce factor is defined by k (number of heads), but not the feature map size or stage number.
3) The authors propose several modifications about ViT, but most of them are already proposed by several papers. For example, mixing attention map with Conv and normalization is used in DeepViT[1]. The spatial reduction of K and V is already used in PvT[2] and CvT[3]. The DWConv-based position embedding is proposed in CPVT[4]. Overlapped token embedding is also used in CvT[3] and T2T[5]. This limits the contribution of this paper.
4) The experiment is not convincing enough. Although the authors mentioned many ViT-based methods in related works, the experiment part only compared with a few of them. For example, for the ImageNet part, there are many strong methods, such as T2T, CvT. But the authors only compare with DeiT, PvT, and Swin. The detection and segmentation experiment on COCO also misses the comparison with Swin, which is the SOTA method at the NeurIPS submission deadline.
5) The experiment setting on COCO is not fair. The PvT and ResNet use single-scale training but the authors use multi-scale training. So the comparison is totally unfair and does not make sense. Meanwhile, I notice the multi-scale training resizes the short side between 480 and 800, which is the setting from Swin, but not PvT (640 to 800). Why not compare with Swin while using the setting from Swin?
6) The experiment lacks a large model comparison. This paper's ``Large'' model is only 50M and 8G FLOPs, which is quite small. This also makes the experiment results not convincing enough.
7) Some citation is not correct, [7] in the paper is ViT, but not PvT.


Overall, this paper proposes several modifications to ViT and improved its performance. But most of the modifications are already proposed in previous papers, and the experiment lacks comparison, some of the comparisons are even not fair. Due to the above analysis, I give a ``5: Marginally below the acceptance threshold'' to this paper.

[1] Zhou, Daquan, et al. "Deepvit: Towards deeper vision transformer."

[2] Wang, Wenhai, et al. "Pyramid vision transformer: A versatile backbone for dense prediction without convolutions."

[3] Wu, Haiping, et al. "Cvt: Introducing convolutions to vision transformers."

[4] Chu, Xiangxiang, et al. "Conditional positional encodings for vision transformers."

[5] Yuan, Li, et al. "Tokens-to-token vit: Training vision transformers from scratch on imagenet."



**Time Spent Reviewing:**

5

---

> ### Author Response · Authors · 2021-08-10
> **Comparison with existing preprint Vision Transformers, setting of reduction stride and experimental results with Swin are provided.**
>
> **Q1**: The analysis of the second shortcoming of ViT is weird. The authors think the multi-head with a low subset head dimension may influence the performance. But no experimental results are provided to prove it. It also contradicts with common sense that more head brings better model capability.
>
> **A1**: Here, we give a detailed analysis of the second shortcoming of ViT, which will be included in the revised version, if necessary. Let the size of input x be $n \times d_m$, where n is the spatial dim and $d_m$ is the channel dim. In MSA, x is split into k heads, each with size $n \times d_k$, where $d_k = \frac{d_m}{k}$. After projection, the attention score $A_i$ of head\_i is calculated as
>
> $ A_i = {\rm Softmax} (\frac{Q_iK_i^{\rm T}}{\sqrt{d_k}}) $
>
> Where $A_i \in R^{n\times n}$, $Q_i, K_i \in R^{n \times d_k}$. Softmax function and $\sqrt{d_k}$ are scaling methods. Therefore, we only consider the capability of $ Q_i K_i^{\rm T}$.
>
> The process to get $A_i$ can be seen as Low-rank Matrix Factorization [3]. Therefore, higher $d_k$ of $Q_i$ and $K_i$ can bring better capability of $A_i$. Also, if $A_i$ is fixed, higher k is better. However, $d_k = \frac{d_m}{k}$, which means, if k is higher, $d_k$ becomes lower, capability of $A_i$ is lower, which may lead to worse capability of MSA(all $k$ heads). To this end, for a given input ($d_m$ is fixed), setting k properly is necessary for MSA. In Transformer [1], given $d_m=512$, k is set as 8, with $d_k$=64, not the maximum 512. This setting is also adopted by ViT [3], and Swin [4], etc. In ResT, we keep $d_k=64$ or $d_k=96$ for different stages, i.e., k is proportional to $d_m$.
>
> **Q2**: The definition of s=8/k is weird and does not make sense. Why the spatial reduce factor is defined by k, but not the feature map size or stage number.
>
> **A2**: In fact, spatial reduction defined by 8/k is the same as defined by the feature map size or the stage number. In this paper, we set k=1, 2, 4, 8 for stage 1, 2, 3, 4. The reduction strides are 4, 8, 16, 32, the same as ResNet. In this setting, k is proportional to reduction strides.
>
> **Q3**: The authors propose several modifications about ViT, but most of them are already proposed by several papers. This limits the contribution of this paper.
>
> **A3**: Thanks for reminding these preprint works, which can be considered as proposed in almost the same period as our ResT. To distinguish our work among them, we compare ResT with them in detail as follows:
>
> (1) ResT vs. DeepViT [5]: Either EMSA in ResT and Re-Attention in DeepViT are the extension of the original MSA module, which validates the contribution of our work from another perspective.
>
> $ EMSA (Q, K, V) ={\rm IN}({\rm Softmax}({\rm Conv}(\frac{QK^{\rm T}}{\sqrt{d_k}})))V $
>
> $ Re-Attention (Q, K, V) ={\rm Norm} ({\rm \Theta^T}({\rm Softmax}(\frac{QK^{\rm T}}{\sqrt{d_k}})))V $
>
> - Norm in Re-Attention is used to reduce the layer-wise variance. Therefore, Norm in Re-Attention is Batch Norm. However, in ResT, Norm is adopted to restore the diversity impaired by the previous Conv, Hence, we use Instance Norm (Experiments are shown in the Appendix section, in Supplementary Material).
>
> - Both EMSA and Re-Attention model cross-head communication. But the difference is adequately clear. $\Theta ^{\rm T} $ in Re-Attention is adopted after Softmax, which changes the distribution of attention score, particularly the variance. Therefore, another scaling function called reatten\_scale (from the official implementation of DeepViT) is used after Norm. However, Conv in EMSA is used before Softmax, which means that there is no need to re-scale the attention score at all.
>
> - Motivation: Re-Attention is proposed to address the attention collapse issue in ViT, in which heads for all blocks are fixed. Nevertheless, EMSA in ResT is proposed to improve the capability of each head, and the number of head changes in different stages. For example, head num is 1 for stage1, and 8 for stage4. Since only several transformer blocks are involved in one stage of ResT, the attention collapse issue is not obvious in ResT.
>
> (2) Spatial reduction in ResT, PVT [6], and CvT [7].
>
> a) SRA in PVT first reshapes the input x with shape $HW \times C$ to the sequence with a size of $\frac{HW}{s^2} \times s^2C$, then a linear projection is adopted to reduce the input dim. Spatial reduction is conducted on K and V.
>
> b) CVT applies Convolutional Projection to Q, K, V to reduce computation cost. The convolution project consists of "Depth-wise Conv2d-> BatchNorm2d-> Point-wise Conv2d" [7].
>
> c) Different from PVT and CvT, in EMSA, the resolution of the input for K and V is directly reduced by an overlapping "Depth-wise Conv2d", which is more efficient than them.
>
> (3) positional Encoding in ResT and CVPT[2].
>
> positional encoding in CVPT is implemented by a Depth-wise Conv2d and is suggested to be added to the output of the first block of transformer encoder. In ResT, positional encoding is built as **Spatial Attention** and is embedded into the patch embedding module. We argue that "PA can be replaced by any spatial attention modules, making the positional encoding flexible in ResT"(Line 167-169). A standard spatial attention contains 3 parts: 1) A transform function: it can be a Conv or more complex module; 2) gated function: usually sigmoid or softmax; 3) dot-product of the input and weights. In ResT, DWConv operation is only the transform function of PA, not the entire PA. Let x be the input token, $\sigma(.)$ be sigmoid function, then the difference of positional encoding in CVPT and ResT can be represented as:
>
> ${\rm CVPT: \hat{x} = x + DWConv(x)}$
>
> ${\rm ResT: \hat{x} = x * \sigma (DWConv(x))}$
>
> (4) Overlapped token embedding is also used in CvT and T2T [8].
>
> Stem is a basic module for both CNN and Transformer backbones, and overlapped embedding is a popular way. For example, Conv7x7 with stride=2, followed by one max-pooling is adopted in ResNet. We discussed the popular stem module in CNN and Transformer, and design a better one for ResT.
>
> **Q4**: More experimental results.
>
> **A4**: Thank you very much for your kind suggestion. We provide the experimental results compared with more backbones here, which will be also included in the revised version.
>
> (1) Compared with more transformer backbones on ImageNet-1k: evaluated on $224 \times 224$ resolution. Inference FPS is measured on a single V100 GPU.
>
> | Methods    | Top-1 | Param. | GFLOPs | FPS |
> |------------|-------|--------|--------|-----|
> | T2T-ViT-14 | 81.5  | 22     | 5.2    | 501 |
> | CVT-13     | 81.6  | 20     | 4.5    | 587 |
> | ResT-Base  | 81.6  | 30     | 4.3    | 673 |
> | T2T-ViT-19 | 81.9  | 39     | 8.9    | 298 |
> | CVT-21     | 82.5  | 32     | 7.1    | 416 |
> | ResT-Large | 83.6  | 51.6   | 7.9    | 429 |
>
> As can be seen, for the smaller model, T2T-ViT-14, CVT-13 and ResT-Base achieve similar top-1 accuracy, but ResT-Base is much faster than T2T-ViT-14 and CVT-13. For the larger model, T2T-ViT-19 is obviously inferior to CVT-21 and ResT-Large, both on accuracy and inference speed. ResT-Large shares a similar speed with CVT-21, and still improves the accuracy by +1.1\%. Therefore, compared with T2T-ViT and CVT, our ResT is a much better backbone.
>
> (2) Object detection performance on COCO val2017 using the RetinaNet framework.
>
> | Backbone   | Schedule | AP          |
> |------------|----------|-------------|
> | R-50       | 1x       | 37.4        |
> | Swin-T     | 1x       | 41.5        |
> | ResT-Base  | 1x       | 42.0 (+4.6) |
> | R-101      | 1x       | 38.5        |
> | Swin-S     | 1x       | 44.5        |
> | ResT-Large | 1x       | 44.8 (+6.3) |
>
> As can be seen, ResT achieves slightly better performance over Swin on the detection task.
>
> (3) Instance Segmentation performance on COCO val2017 using the Mask R-CNN framework.
>
> | Backbone   | Schedule | box AP | mask AP |
> |------------|----------|--------|---------|
> | R-50       | 1x       | 38.0   | 34.4    |
> | Swin-T     | 1x       | 42.2   | 39.1    |
> | ResT-Base  | 1x       | 41.6   | 38.7    |
> | R-101      | 1x       | 40.4   | 36.4    |
> | Swin-S     | 1x       | 44.8   | 40.9    |
> | ResT-Large | 1x       | 44.5   | 40.7    |
>
> Results show that ResT is competitive to Swin Transformer on the instance segmentation.
>
> **Conclusion**: ResT, and Swin, are both preprint manuscripts at the same period. Both explore the design of ViT (from different perspectives) and achieve similar results with similar model complexity. To this end, it is safe to say the proposed ResT is a strong competitor of Swin.
>
> **Q5**: This paper's "Large" model is only 50M and 8G FLOPs, which is quite small. This also makes the experiment results not convincing enough.
>
> **A5**: Thanks for pointing out that. We do take it as an advantage of ResT. A more lightweight and affordable model is believed to bring much more benefits to the whole research community.
>
> **Q6**: Some citation is not correct, "[7]" in the paper is ViT, but not PVT.
>
> **A6**: Thank you very much for your help. This mistake will be corrected in the revised version.
>
> [1] Vaswani, Ashish, et al. "Attention is all you need". NeurIPS 2017
>
> [2] Chu, Xiangxiang, et al. "Conditional positional encodings for vision transformers." arXiv preprint arXiv:2102.10882
>
> [3] Geng, Zhengyang, et al. "Is Attention Better Than Matrix Decomposition?" ICLR2020
>
> [4] Liu, Ze, et al. "Swin transformer: Hierarchical vision transformer using shifted windows." arXiv preprint arXiv:2103.14030
>
> [5] Zhou, Daquan, et al. "Deepvit: Towards deeper vision transformer." arXiv preprint arXiv:2103.11886
>
> [6] Wang, Wenhai, et al. "Pyramid vision transformer: A versatile backbone for dense prediction without convolutions. "arXiv preprint arXiv:2102.12122
>
> [7] Wu, Haiping, et al. "Cvt: Introducing convolutions to vision transformers." arXiv preprint arXiv:2103.15808
>
> [8] Yuan, Li, et al. "Tokens-to-token vit: Training vision transformers from scratch on imagenet." arXiv preprint arXiv:2101.11986

---

### Official Review · Reviewer_Qyop · 2021-07-14

**Rating:** 6
**Confidence:** 5

**Summary:**

This paper presents an efficient multi-head self-attention by reducing the tokens of keys and values; furthermore, this paper also proposed another way to encode position information in the ViT. The authors validated their methods in image classification and object detection.

**Limitations And Societal Impact:**

Yes

**Main Review:**


1. This paper improves the vanilla ViT by integrating many useful techniques, e.g. pyramid structure, EMSA, PA, etc.

2.  The proposed components are very similar to the previous work, making the novelty of this paper limited.

    - EMSA, it is similar to PVT. Except for the authors use depthwise convolution to reduce spatial tokens; and the instance normalization and 1d Conv on the attention map. Does IN and 1d conv play an important role for the large model?

    - Using a stronger patch embedding could lead to better performance had been mentioned in the original ViT paper for small-scale data, e.g. ImageNet1K.

Questions

1. See point 2 above

2. Why do the authors use stronger data augmentation (multi-scale) in object detection? in PVT, for 1x schedule, they do not use multi-scale training. What is the performance if disabling multi-scale training?
Furthermore, why authors do not compare to Swin in the detection as they are in the classification.
(Others reproduce Swin results with ImageNet1K pretrained weights. "Twins: Revisiting the Design of Spatial Attention in Vision Transformers.")

3. Why the ablation study does not use the same settings, do those observations still hold if using the same settings? Especially, why only use simpler data augmentation?



**Time Spent Reviewing:**

8

---

> ### Author Response · Authors · 2021-08-10
> **Comparison with PVT and Swin Transformer, and the explanation of settings in the ablation study are provided.**
>
> Thank you very much for the insightful and constructive comments. Codes and Appendix of this paper are available in the Supplementary Materials, which contains more contents helping for the understanding of this manuscript.
>
> **Q1**: EMSA is similar to PVT, except for the authors use depth-wise convolution to reduce spatial tokens; and the instance normalization and 1d Conv on the attention map. Does IN and 1d Conv play an important role for the large model?
>
> **A1**: (1) SRA in PVT [1] and EMSA in the proposed ResT are both used to reduce the computation cost for high-resolution feature maps. The followings are the difference:
>
> - SRA first reshapes the input x with shape $\mathbb{R}^{HW \times C}$ to the sequence with a size of $\mathbb{R}^{\frac{HW}{s^2} \times s^2 C}$, then a linear projection is adopted to reduce the dimension of the input sequence [1].
>
> - In EMSA, the resolution of the input for key and value is directly reduced by an overlapping Depth-wise Conv2d, which is more efficient and can keep the position information of the original input. In addition, EMSA also models the interaction across the attention-heads dimension while keeping the ability of diversity of multi-heads. This is of great importance, particularly when the tokens embedding dimension (for each head) is short. We demonstrate this in the Ablation Study section and the Appendix section (in the Supplementary Materials).
>
> (2) IN (short for Instance Normalization) and Conv 1x1 in EMSA still play an important role for the large model. The Top-1 Accuracy of ResT-Large with Conv 1x1 and IN (i.e., setting "apply\_transform=True" in the "rest.py", which is available at the Supplementary Materials) still improved by +0.59 compared with that without Conv 1x1 and IN (i.e., setting "apply\_transform=False"). As shown in Table 1 of the paper, the number of heads is increased as the embedding dimension (i.e., $d_m$ in Section 2.2) of input tokens increase. As a result, each head is responsible for a fixed number (64 or 96) of the embedding dimension of the input tokens in different stages. In this setting, Conv 1x1 and IN are very important.
>
> **Q2**: Using a stronger patch embedding could lead to better performance had been mentioned in the original ViT paper for small-scale data, e.g. ImageNet1K.
>
> **A2**: ViT mentioned "as an alternative to raw image patches, the input sequence can be formed from feature maps of a CNN"[5], and call this a hybrid model. But providing no further discussion about CNN stem. In our paper, we discussed and compared the commonly used ResNet stem module, i.e., Conv7x7 with stride 2, following by a max-pooling operation, and the original ViT stem (i.e., patch embedding). We also proposed a new stem module, i.e., stacking of 3 overlapping Conv3x3 layers. In addition, unlike ViT, which only adopting one patch embedding module, our ResT is constructed as an efficient multi-scale vision Transformer. Therefore, four patch embedding modules are used in ResT. The main idea of patch embedding of ResT is hierarchically expanded the channel capacity while reducing the spatial resolution with overlapping convolution operations. Therefore, the patch embedding module is substantially different from the original one and the hybrid one in ViT.
>
> **Q3**: Why do the authors use stronger data augmentation (multi-scale) in object detection?
>
> **A3**: We followed the same setting as Swin Transformer [2] in object detection. The comparisons between Swin (Results are reproduced by Twins [3]) and ResT on the COCO val2017 using the RetinaNet framework are as follows:
>
> | Methods   | Backbone   | Schedule | AP          |
> |-----------|------------|----------|-------------|
> | RetinaNet | R-50       | 1x       | 37.4        |
> | RetinaNet | Swin-T     | 1x       | 41.5        |
> | RetinaNet | ResT-Base  | 1x       | 42.0 (+4.6) |
> | RetinaNet | R-101      | 1x       | 38.5        |
> | RetinaNet | Swin-S     | 1x       | 44.5        |
> | RetinaNet | ResT-Large | 1x       | 44.8 (+6.3) |
>
> As can be seen from the above results, ResT achieves slightly better performance over Swin on the detection task.
>
> **Q4**: Why the ablation study does not use the same settings, do those observations still hold if using the same settings? Especially, why only use simpler data augmentation?
>
> **A4**: We adopt the simplest data augmentation and hyper-parameters settings in ResNet [4] to thoroughly investigate the important components of ResT, i.e., eliminating the influence of strong data augmentation and training tricks. Under this setting, we demonstrate that Vision Transformers can still achieve better results without training tricks. Specifically, the Top-1 accuracy of ResT-Lite is 72.88, which outperforms the ResNet-18 (69.76) by +3.1 improvement. We believe the same setting as ResNet in the ablation study can eliminate the doubt of Vision Transformer to some extent, i.e., the improvements of Vision Transformer over CNN mainly come with strong data augmentation and training tricks. This can promote the ongoing and future research of Vision Transformer.
>
> **Reference**
>
> [1] Wang, Wenhai, et al. "Pyramid vision transformer: A versatile backbone for dense prediction without convolutions."
> arXiv preprint arXiv:2102.12122 (2021).
>
> [2] Liu, Ze, et al. "Swin transformer: Hierarchical vision transformer using shifted windows." arXiv preprint arXiv:
> 2103.14030 (2021).
>
> [3] Chu, Xiangxiang, et al. "Twins: Revisiting the design of spatial attention in vision transformers." arXiv preprint arXiv:2104.13840 1.2 (2021)
>
> [4] He, Kaiming, et al. "Deep residual learning for image recognition". CVPR2016.
>
> [5] Dosovitskiy, Alexey, et al. "An image is worth 16x16 words: Transformers for image recognition at scale". ICLR 2021.

---

> > ### Comment · Reviewer_Qyop · 2021-08-26
> > **Thanks for your rebuttal**
> >
> > The authors addressed most of my concerns.
> > However, I still have the concern of the different setup in the ablation study. E.g., the standard way to train ViT could not get a good result if only ImageNet1K is used. That means the training protocol affects the results. Now, the authors showed the final results with setting A but perform ablation study with setting B. It is hard to clearly understand how much gain comes from the proposed method.
> >
> > I upgraded my score to 6.

---

> > > ### Author Response · Authors · 2021-08-27
> > > **More clarifications of ablation study settings**
> > >
> > > Thank you very much for the constructive feedback.
> > >
> > > The main concern for adopting the current ablation study setting
> > > is to thoroughly investigate the critical components of the proposed ResT, which is totally different from ViT (e.g.,
> > > the entire pipeline, the patch embedding way, the multi-head self-attention module, and the positional encoding method).
> > > Therefore, we just adopted the simplest data augmentation and hyper-parameters settings to eliminate any influence that
> > > might be brought by strong data augmentation and training tricks.
> > >
> > > In addition, I do agree that the training protocol affects the results. However, the effects are the same for models
> > > with similar structures (e.g., models in ablation study). Here, we provide the ImageNet-1k classification results with
> > > different parameters reducing methods of EMSA (in Figure 3) in ResT-Lite as an explanation.
> > >
> > > | Reduction Methods | Ablation Setting |            | Main Setting |            |
> > > |-------------------|-------------------------|------------|--------------|------------|
> > > |                   | Top-1 Acc.              | Top-5 Acc. | Top-1 Acc.   | Top-5 Acc. |
> > > | DWConv            | 72.88                   | 90.62      | 77.20        | 93.71      |
> > > | Average Pooling   | 72.64                   | 90.41      | 77.03        | 93.58      |
> > > | Max Pooling       | 72.20                   | 89.97      | 76.95        | 93.40      |
> > >
> > > We can see that the experiment's settings significantly affect the absolute results of different types of ResT-Lite.
> > > However, the relative rankings remain unchanged, i.e., DWConv > Average Pooling > Max Pooling.
> > >
> > > Finally, for models with totally different structures, such as the proposed ResT and ViT, we compare their results in
> > > the same setting, showing in Table 2.

---

> > > > ### Comment · Reviewer_Qyop · 2021-09-02
> > > > **Response**
> > > >
> > > > Thanks for your follow-up rebuttal. As you show here, with the main setting, even though the relative order still holds but the gain becomes much smaller (0.68% -> 0.25%, DWconv vs. Max.).

---

### Official Review · Reviewer_Wueu · 2021-07-16

**Rating:** 6
**Confidence:** 4

**Summary:**

This work presents a computationally efficient vision transformer network for image recognition. In particular, this work propose memory-efficient self-attention module to replace the original one in the original transformer. Second, it modifies the inflexible position encoding (that only allows fixed length of tokens) to accept variable size of input images. Lastly, the patch embedding is achieved by stacking overlapping convolution operations with strides. Extensive experiments are conducted to show the superiority of the proposed ResT.

**Limitations And Societal Impact:**

Yes.

**Main Review:**

Pros:
- The writing is good. The paper is easy to follow.
- The proposed methods are intuitive and reasonable, including efficient transformer block and the flexible position encoding.
- The proposed ResT is a general backbone and is expected to benefit many vision tasks.
- The code of this work is provided.

Cons:
- The proposed methods seem to apply some CNN components/designs (eg. depthwise convolution and stem) to the transformers, which might not be considered as major novelty and is more like incremental improvements over ViT.
- The abstract says ResT is a general-purpose backbone, so I would expect some experiments on one or two more vision tasks, such as semantic segmentation or pose estimation.
- The improvements over Swin transformer in Table 2 are not so obvious.
- Can the author provide some detection results on Swin transformer on COCO??

**Time Spent Reviewing:**

1.5

---

> ### Author Response · Authors · 2021-08-10
> **Novelty besides applying CNN compenents is provided. Also experiments results compared with Swin on semantic segmentation and object detection are given.**
>
> Thank you very much for the insightful and constructive comments. Codes and Appendix of this paper are available in the Supplementary Materials, which contains more contents helping for the understanding of this manuscript.
>
> **Q1**: The proposed methods seem to apply some CNN components/designs (e.g., Depth-wise convolution and stem) to the transformers, which might not be considered a major novelty and is more like incremental improvements over ViT.
>
> **A1**: Besides applying CNN components/designs to the transformer, the proposed ResT still has the following three advantages over the origin ViT:
>
> (1) In ResT, four patch embedding modules are applied, which create a multi-scale pyramid of features by hierarchically expanding the channel capacity while reducing the spatial resolution. As a result, ResT is constructed as a multi-scale transformer and can be served as a general-purpose backbone for dense predictions.
>
> (2) Computation costs are significantly reduced in EMSA, particularly in the earlier stages, where the spatial dimension of the input token is high. Only about $1/s^2$ (s is the reduction factor) computation costs of MSA are included in the EMSA. Besides, cross-head information is modeled in EMSA to improve the capability of each head while keeping the ability of diversity of multi-heads.
>
> (3) positional encoding is constructed as spatial attention and embedded into the patch embedding module, making the proposed ResT easier to deploy and can flexibly tackle input images with arbitrary size.
>
> To summarize，the proposed ResT provides new insights into designing vision transformers, which is expected to benefit and promote further research of vision transformer.
>
> **Q2**: The abstract says ResT is a general-purpose backbone, so I would expect some experiments on one or two more vision tasks.
>
> **A2**: Thank you very much for your constructive suggestion. We provide semantic segmentation results on ADE20K here, which will be also included in the revised version. Specifically, we utilize UperNet [1] in mmsegmentation [2] as our base framework for its high efficiency and follow the same setting as Swin [3]. The followings are the comparisons with different backbones on the ADE20K validation dataset.
>
> | Method  | Backbone   | mIoU        |
> |---------|------------|-------------|
> | UperNet | R-50       | 42.1        |
> | UperNet | Swin-T     | 44.4        |
> | UperNet | ResT-Base  | 44.6 (+2.5) |
> | UperNet | R-101      | 43.8        |
> | UperNet | Swin-S     | 47.7        |
> | UperNet | ResT-Large | 47.8 (+4.0) |
>
> We can see, ResT is able to achieve competitive mIoU to Swin Transformer.
>
> **Q3**: The improvements over Swin Transformer in Table 2 are not so obvious.
>
> **A3**: First, compared to Swin backbones with similar model complexities(i.e., Swin-T, Swin-S), the proposed ResT (ResT-Base, ResT-Large) achieve a higher Top-1 accuracy (+0.3). In addition, ResT provides a lightweight model, i.e., ResT-Lite, which achieves 77.2 Top-1 accuracy with only 1.4M parameters and 10.5 GFlops, making it suitable for deploying in real-time applications, especially on devices with limited resources, such as autonomous driving cars. A lightweight and affordable model is believed to bring much more benefits to the whole research community.
>
> **Q4**: Can the author provide some detection results on Swin transformer on COCO?
>
> **A4**: Thank you very much for your constructive suggestion. We provide the object detection results on the COCO val2017 using the RetinaNet framework here, which will be included in the revised version.
>
> | Methods   | Backbone   | Schedule | AP          |
> |-----------|------------|----------|-------------|
> | RetinaNet | R-50       | 1x       | 37.4        |
> | RetinaNet | Swin-T     | 1x       | 41.5        |
> | RetinaNet | ResT-Base  | 1x       | 42.0 (+4.6) |
> | RetinaNet | R-101      | 1x       | 38.5        |
> | RetinaNet | Swin-S     | 1x       | 44.5        |
> | RetinaNet | ResT-Large | 1x       | 44.8 (+6.3) |
>
> As can be seen from the results, ResT achieves slightly better performance over Swin on the detection task.
>
> **Conclusion**: The proposed ResT, and Swin Transformer, both have preprint versions available online during this year's NeurIPS submission period. They both explore the design of ViT (from different perspectives). In addition, ResT can tackle input images with arbitrary size without interpolation or fine-tuning, which is a great advantage over Swin, especially on downstream tasks. Therefore, we can say our proposed ResT is very competitive to Swin.
>
> **Reference**
>
> [1] Xiao, Tete, et al. "Unified perceptual parsing for scene understanding". ECCV 2018.
>
> [2] Contributors, MMSegmentation. "MMSegmentation: Openmmlab semantic segmentation toolbox and benchmark." (2020).
>
> [3] Liu, Ze, et al. "Swin transformer: Hierarchical vision transformer using shifted windows." arXiv preprint arXiv:2103.14030 (2021).

---

### Official Review · Reviewer_3XRr · 2021-07-16

**Rating:** 6
**Confidence:** 4

**Summary:**

The authors propose a new ViT-based architecture for image recognition. The architecture is based on 3 main ideas: 1) using channel-wise conv to reduce the spatial dimension of feature maps to save computations. 2) using a stack of convs as "patch embedding", which is used at the beginning of each stage to shrink spatial size while increasing channel dimensions. 3) A depth-wise conv to replace the "position encoding" in the original ViT.

Overall the model achieves strong performance, outperforming Swin-Transformer at comparable computational cost.



**Limitations And Societal Impact:**

I've listed the limitations in my review above. I don't see anything particularly concerning regarding societal impact, compared to other deep network architecture.

**Main Review:**

In the following I review this paper in the following axes.

[originality: medium]
 - Reducing the spatial dimension to save computational costs and memory has been used in, e.g., "non-local neural networks" (Wang et al., CVPR 18).

 - Using convs with stride to reduce spatial dimensions and increase channel dimensions has been the standard practice in many conv nets (e.g., ResNet).

 - Replacing positional encoding with the depth-wise conv is original/novel to my knowledge. I find this part quite interesting.

[quality: medium-high]
 - The experiments are satisfactory, covering important ablation studies that I'd expect. The results are impressive --- high performance with low computational costs compared to recent, strong ViT-based methods.

[clarity: low-medium]
I find some descriptions of this paper confusing:
1. L159: The description of "GL" is vague. Should be mathematically defined.
2. "pixel-wise attention (PA)" turns out to be a "conv" operator. Something like "conv-based re-weighting" might be clearer.
3. I found 2.4 confusing. If I understand it correctly, in (7) GL(x) operates on features "x", instead of the "positions". Thus this operator doesn't explicitly encode any position. Why is it then called a "positional encoding"? (I understand that if we implement GL as conv (PA), it can model some spatial layout to a certain extent, but this is not positional encoding. This is using conv nets to model spatial patterns.)

[significance: medium]
 - The area of ViT based models have many interesting open questions to be explored. This work advances the model design through solid improvement in accuracy. I think this work can be useful for future research.


**Time Spent Reviewing:**

1.5 hours

---

> ### Author Response · Authors · 2021-08-10
> **Mathematically defined of GL, why it is being called as "positional encoding", and the difference between PA and Conv are given.**
>
> Thank you very much for the insightful and constructive comments. Codes and Appendix of this paper are available in the Supplementary Materials, which contains more contents helping for the understanding of this manuscript.
>
> **Q1**: The description of "GL" is vague. Should be mathematically defined.
>
> **A1**: GL is the short of ''Group Linear'', which can be viewed as a group Conv 1x1 operation with group g. It can be defined as follows:
>
> Given x with size $n\times d_m$, where n is spatial dimension and $d_m$ is the channel dimension, GL first splits x into g non-overlapping groups such that $x = Concat (x_1, \cdots, x_g)$, where the size of $x_i$ is $n \times \frac{d_m}{g}$. all $x_i$’s are then simultaneously transformed by g linear operations to produce g outputs $y_i = x_i W_i$, where $W_i$ is the linear operation weight. $y_i$’s are then concatenated to produce the final $n\times d_m$ output $y = Concat(y1, \cdots, y_g)$. In ResT, we set $g= d_m$, i.e., the channel dimension of the input.
>
> **Q2**: "pixel-wise attention (PA)" turns out to be a "conv" operator. Something like "conv-based re-weighting" might be clearer.
>
> **A2**:Yes, PA is a "conv-based re-weighting" operation. Let x be the input, PA contains three procedures: (1) using DWConv to obtain pixel-wise weight of x, i.e., $ {\rm x1 = DWConv(x)}$; (2) a gated function (e.g., sigmoid) is adopted to enable guidance for precise and adaptive selection, i.e., $ {\rm x2 = \sigma(x1)}$; (3) re-weighting to the input: ${\rm PA(x) = x * x2}$. Therefore, PA is also a standard spatial attention operation, while the DWConv operation is only part of PA. In addition, we point out that "PA can be replaced by any spatial attention modules, making the positional encoding flexible in ResT"(Line 167-169 in section 2.4).
>
> **Q3**: Why is GL(x) being called a "positional encoding"?
>
> **A3**: GL(x) is related to x and only re-weights the values of x, not relevant to absolute or relative position information. Therefore, GL(x) is not a strict "positional encoding". However, for a specific input x, ${\rm x+\theta}$ may equal to ${\rm x + GL(x)}$, since both of $\theta$ and GL(x) involve learnable parameters to re-weight the values. We first explore GL(x) to see whether we can dynamically generate positional encoding according to the input. Ablation study results show that adopting $\theta$ and GL(x) as positional encoding can achieve similar results. This observation suggests that positional encoding can be conditioned to the input. Inspired by this, we propose to construct positional encoding as spatial attention (e.g., PA), which is a strict "positional encoding" since relative position information is added for the input. For the convenience of description, we refer to GL(x) as positional encoding in this paper.

---

> > ### Comment · Reviewer_3XRr · 2021-08-24
> > **Re: Mathematically defined of GL, why it is being called as "positional encoding", and the difference between PA and Conv are given.**
> >
> > Thanks for the clarification. I think adding these discussions to the paper would be helpful.
> >
> > After reading other reviews and the author responses, I still think that although the techniques explored are not entirely new, this paper presents extensive empirical results that can be useful reference for future research. Overall I keep my final recommendation unchanged.

---

> > > ### Author Response · Authors · 2021-08-27
> > > **Thanks for the feedback**
> > >
> > > We greatly appreciate your precious feedback for our research. The mathematically defined of GL will be included in the final version.

---

### Decision · Program_Chairs · 2021-09-27

**Decision:**

Accept (Poster)

**Comment:**

Three reviewers recommend borderline-acceptance, one reviewer recommends borderline-rejection. Most remaining concerns stem from a lack of novelty compared to a very rapidly moving (partially concurrent) work on vision transformers. However, the AC agrees with the majority of reviewers that there is enough merit in the presented work to justify acceptance.